# SYNBUILD-3D: A MULTI-MODAL SYNTHETIC DATASET OF OVER 100,000 SEMANTICALLY ENRICHED 3D BUILDING WIREFRAMES WITH AI-GENERATED FLOOR PLANS

## ABSTRACT

Modeling precise geometric and semantic relationships in 3D remains one of the greatest challenges in generative machine learning today, partly because of a lack of large 3D datasets in the public domain. Drawing upon the successful adoption of synthetic datasets in the computer vision community, we propose to address this challenge in the context of 3D buildings with SYNBUILD-3D, a large, multi-modal, and domain-specific dataset of more than 100,000 3D building wireframes along with their corresponding floor plan images. Unlike existing 3D building datasets, SYNBUILD-3D has been designed with and validated by building modeling and simulation experts, providing rich geometric and semantic information. As a result, SYNBUILD-3D is, to the best of our knowledge, the first 3D building dataset that provides interior and exterior building geometries, including the position and size of doors and windows derived from the floor plans. By releasing SYNBUILD-3D, we aim to offer the geometric deep learning community a high-quality dataset for conditional and unconditional 3D building generation tasks. In contrast to existing datasets that typically focus on modeling either the interior or exterior of 3D objects, SYNBUILD-3D can facilitate the development of generative algorithms that account for both perspectives while incorporating geometric and semantic constraints. The dataset and its associated codebase are available at GITHUB LINK.

## 1 INTRODUCTION

Graph-based data representations play a crucial role in fields like computer graphics, molecule generation, and architecture, thanks to their ability to define complex geometric and semantic features with precision and efficiency. Yet, despite rapid advances in generative modeling, generating 3D graphs remains a major challenge. In the context of 3D buildings, this is partly because buildings represented as wireframe graphs exhibit intricate incident relationships and connectivity constraints, requiring a more complex and holistic spatial understanding than the generation of images, natural language, or solid 3D meshes. However, 3D building wireframe datasets are rare and tend to be proprietary or simplistic in their geometric and semantic structure. Additionally, most existing datasets focus exclusively on outdoor building environments (Wang et al., 2023; New York City Department of City Planning, 2014). Inspired by the progress sparked by large graph-based datasets in the molecule generation field (Ramakrishnan et al., 2014; Axelrod & Gómez-Bombarelli, 2022), we argue that more domain-specific 3D graph datasets are needed.

### 1.1 CONTRIBUTION

To support the development of geometry-aware generative algorithms in the context of 3D buildings, we introduce SYNBUILD-3D along with the following contributions:

1. SYNBUILD-3D provides a large-scale dataset of more than 100,000 semantically enriched 3D building wireframe models along with their respective floor plan images and segmentation masks. Unlike existing datasets, our dataset has been designed with and validated by

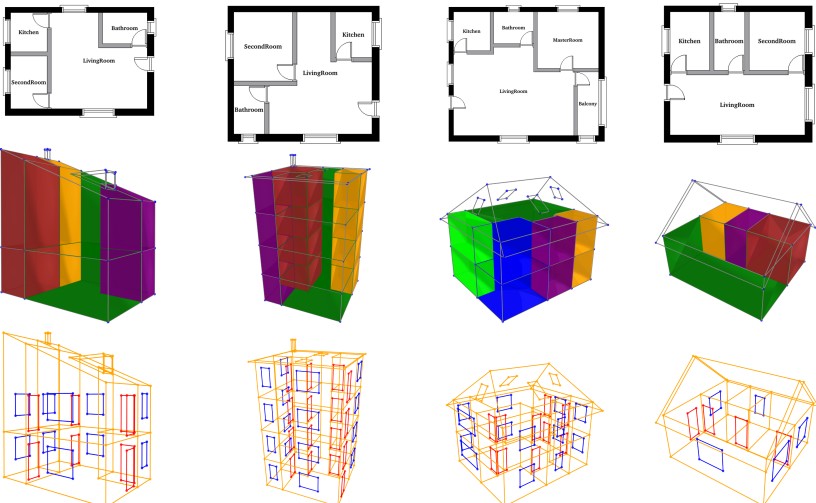

Figure 1: SYNBUILD-3D consists of 103,233 3D building wireframes, floor plan images, and floor plan segmentation masks. Based on the information in each floor plan image (top row), we semantically enrich the corresponding 3D wireframe with information on its room types (middle row) and the position of its doors and windows (bottom row).

3D building modeling and simulation experts. Hence, SYNBUILD-3D represents the interior and exterior structure of each building in a unified wireframe model. Apart from providing precise geometric relationships between different building elements, SYNBUILD-3D includes rich semantic information derived from floor plans such as room types as well as the size and position of doors and windows as illustrated in Figure 1.

2. SYNBUILD-3D provides a multi-modal 3D building dataset. Apart from the semantically enriched 3D wireframe model, each building comes with its floor plan image and its floor plan segmentation mask. As such, SYNBUILD-3D can enable unconditional as well as conditional generation tasks across different data modalities.

3. Along with the dataset, we publish the codebase to produce SYNBUILD-3D. While the dataset demonstrates that the codebase can generate semantically enriched 3D building wireframe models at scale, it is also modular. As a result, the pipeline is flexible and can integrate more advanced building generation modules in the future as they become available.

## 2 RELATED WORK

The subsequent literature review focuses only on publicly available graph-based datasets for 3D objects. After reviewing the current state of mesh-based datasets, we dive deeper into domain-specific datasets in the molecule and 3D building field.

### 2.1 GENERAL MESH DATASETS

As one of the pioneering datasets, ShapeNet has enabled significant progress in the geometric deep learning field, particularly across tasks such as shape recognition and generation (Chang et al., 2015). Providing textured Computer-Aided Design (CAD) models labeled with semantic categories from WordNet (Fellbaum, 1998), it comprises, in theory, more than three million shapes across 3,000 categories. In practice, only around 51,000 models remain when filtering by mesh and texture quality as objects in ShapeNet are affected by their low resolution and overly simplistic textures. Other prominent CAD-based datasets include ModelNet with around 130,000 objects and ABC with over 1 million objects, mostly covering mechanical parts with sharp edges and well-defined surfaces (Wu et al., 2015; Koch et al., 2019). Bringing together 3D objects from various sources such as GitHub, Thingiverse, and Sketchfab, Objaverse-XL is the largest publicly available dataset of 3D

Table 1: Comparison of Graph-based 3D Building Datasets

| | Model Count | Modality | | | | Scope | | Type | | Semantics | | | |
| --- | --- | --- | --- | --- | --- | --- | --- | --- | --- | --- | --- | --- | --- |
| | | Wireframe | Mesh | Image | Point Cloud | Interior | Exterior | Synthetic | Real | Door | Windows | Rooms | |
| Building3D | 160K+/47K+ | ✓ | ✓ | - | ✓ | - | ✓ | - | ✓ | - | - | - | Roof Reconstruction |
| BuildingNet | 2K | - | ✓ | - | ✓ | - | ✓ | ✓ | - | ✓ | ✓ | - | 3D Segmentation |
| CityGML NYC | 1M+ | ✓ | - | - | - | - | ✓ | - | ✓ | - | - | - | Urban Modeling |
| 3D House Wireframe | 79K | ✓ | - | ✓ | - | ✓ | ✓ | ✓ | - | - | - | ✓ | Wireframe Generation |
| SYNBUILD-3D (Ours) | 100K+ | ✓ | - | ✓ | - | ✓ | ✓ | ✓ | - | ✓ | ✓ | ✓ | Wireframe Generation |

objects with more than 10 million unique objects covering a broad range of categories (Deitke et al., 2023). In its current form, Objaverse-XL is an order of magnitude larger than other 3D datasets and significantly improves the zero-shot generalization performance of generative models such as Zero123-XL (Liu et al., 2023). While Objaverse-XL and other prominent 3D mesh-based datasets do contain some 3D building models and 3D building components, these objects are modeled as solid meshes. Consequently, these datasets are of limited use when it comes to obtaining building wireframes that account for a building's internal and external structure at the same time.

## 2.2 MOLECULE DATASETS

In molecular structure research, datasets like QM9 and GEOM have been pivotal in advancing the development of domain-specific algorithms for generating 3D molecular graphs (Ramakrishnan et al., 2014; Axelrod & Gómez-Bombarelli, 2022), as demonstrated by recent advances such as (Xu et al., 2022; 2023; Satorras et al., 2022; Vignac et al., 2023). While datasets like QM9 and GEOM are a rich source of atomic and spatial data, molecular structures are inherently defined by a set of physical rules on atomic attraction and interaction. Hence, molecular structures exhibit designs and geometric constraints that are different from many man-made objects where aesthetics and functional requirements play an important role. In contrast to man-made objects such as buildings or cars, it also does not necessarily make sense to distinguish between a molecule's interior and exterior structure.

## 2.3 3D BUILDING DATASETS

In the context of 3D buildings, relevant graph datasets include Building3D (Wang et al., 2023), BuildingNet (Selvaraju et al., 2021), CityGML-based datasets, e.g., (New York City Department of City Planning, 2014), and the 3D House Wireframe dataset (Ma et al., 2024) as illustrated in Table 1. Building3D is a dataset of over 160,000 building point clouds derived from airborne LiDAR. It is geared towards reconstructing roofs in 3D, but provides simplified building meshes without semantics for around 47,000 building exteriors as well as roof wireframes. While valuable for roof reconstruction tasks, Building3D lacks interior structural information and detailed semantic annotations beyond the roof. BuildingNet focuses on 3D semantic segmentation and offers granular semantic information for around 2,000 3D building exteriors. While it provides rich hierarchical information on individual building parts, its applicability for training generative algorithms is limited due to its small size. CityGML-based datasets, such as the New York City 3D Model, provide detailed 3D building models at scale, i.e., over one million buildings in the case of New York City. Going beyond simple geometry, CityGML-based datasets often provide structured information on building attributes like type, height, and land use. However, similar to Building3D and BuildingNet, these datasets do generally also not provide information on building interiors. In comparison to the previously mentioned studies, Ma et al. (2024) presents a more similar approach to ours, positioning itself as the first 3D building wireframe dataset to incorporate both interior and exterior structures. Using floor plan images from the RPLAN dataset (Wu et al., 2019), the authors vectorize and extrude the floor plan walls, before completing the 3D model with an ad-hoc roof wireframe derived via a straight skeleton algorithm. The final dataset consists of 78,791 3D buildings where each building

is composed of three distinct wireframes, i.e., one for the rooms, the exterior walls, and the roof. While the dataset makes use of floor plan semantics such as room types, it does not account for the position and size of doors and windows. Moreover, the ad-hoc algorithm to produce roof wireframes is prone to create unrealistic geometries.

To conclude, existing graph datasets for 3D buildings often focus on building exteriors, lack unified wireframe representations, or do not provide the semantic richness required for advanced generative tasks in architecture and building design. SYNBUILD-3D aims to address this gap by providing a comprehensive dataset of over 100,000 semantically enriched 3D building wireframe models, complete with corresponding floor plan images and segmentation masks. By offering a unified representation of interior and exterior building structures, along with crucial semantic information, SYNBUILD-3D is the first dataset to follow industry standards in the building modeling domain. As a result, SYNBUILD-3D has the potential to significantly advance the development of geometry-aware generative algorithms in the context of 3D buildings.

## 3 DATASET DESCRIPTION

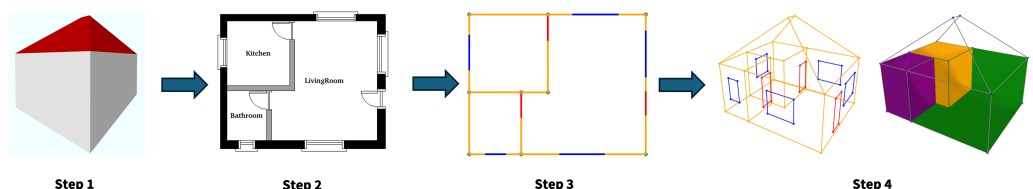

Figure 2: SYNBUILD-3D pipeline. In the first step, a procedural generation engine creates a randomized building exterior. We then use that building's footprint to produce a building-specific floor plan image (Step 2). Afterward, the floor plan information is vectorized (Step 3) and extruded within the building hull (Step 4).

### 3.1 DATASET CREATION

We have developed a multi-step pipeline to generate a dataset of semantically enriched 3D building wireframes with accompanying floor plan images, as illustrated in Figure 2. The subsequent sections provide a detailed explanation of each step in the pipeline.

**Step 1: Procedurally generate randomized building exterior.** In the first step, we use the procedural generation engine described in Biljecki et al. (2016) to generate randomized building exteriors. To ensure the buildings are geometrically sound, the generation process adheres to pre-defined rules while incorporating randomized decisions. For example, Biljecki et al. (2016) creates building exteriors by making randomized decisions in terms of the footprint size, the number of stories, the roof type, and roof superstructures such as dormers, windows, and chimneys. In particular, the procedural generation engine supports five distinct roof types, i.e., flat, gabled, hipped, pyramidal, and shed. After post-processing, the first step yields a wireframe model for each building's exterior.

**Step 2: Generate footprint-conditioned floor plan image.** In the second step, each building's footprint is used to condition an AI-based floor plan generator as described in Wu et al. (2019). Hence, the input to Step 2 is a building footprint boundary, and the output is the corresponding floor plan image along with its segmentation mask. Since the floor plan generator produces different layouts based on the front door position and the footprint area, each building's front door is placed randomly on the footprint boundary.

**Step 3: Vectorize floor plan information.** In the third step, we vectorize the generated floor plan image and enrich the vectorization output with floor plan semantics, such as information about the respective room types and the position and size of doors and windows.

**Step 4: Align and extrude vectorized floor plan within building hull.** Lastly, we align the vectorized and enriched floor plan with the original footprint polygon and extrude it within the building volume produced in Step 1. During the alignment step, we represent both the original footprint poly-

gon and the floor plan as bitmaps such that pixels inside the building are set to one and pixels outside the building are set to zero. We then set up an optimization problem to determine the best parameters for translating $(t_x, t_y)$ and scaling $(s_x, s_y)$ the floor plan bitmap along the x- and y-axis on top of the footprint bitmap. To do so, we compare the transformed floor plan bitmap to the original footprint bitmap across all pixels and minimize the following loss:

$$
\begin{aligned}
\text{Alignment loss}(t_x, t_y, s_x, s_y) &= 20 \cdot \text{Coverage} + \text{Overhang} \\
&= \sum_{i,j} 20 \cdot \mathbb{1}_{FootprintBitmap_{ij} > FloorPlanBitmap(t_x, t_y, s_x, s_y)_{ij}} \\
&\quad + \mathbb{1}_{FloorPlanBitmap(t_x, t_y, s_x, s_y)_{ij} > FootprintBitmap_{ij}}
\end{aligned} \tag{1}
$$

where *Coverage* refers to the total number of pixels where the footprint bitmap extends beyond the transformed floor plan bitmap, while *Overhang* represents the total number of pixels where the transformed floor plan bitmap exceeds the footprint bitmap area.

## 3.2 TECHNICAL VALIDATION AND DATA REPRESENTATION

Following the advice of multiple experts in the building modeling and simulation domain, each building combines interior as well as exterior geometries and semantics in a single wireframe representation. In other words, following EnergyPlus's 3D building modeling paradigm (National Renewable Energy Laboratory, 2017), we represent surfaces such as walls as single planes with no thickness. In the case of EnergyPlus, building surfaces can be enriched with semantic meta-information such as materials, thermal transmittance, and thickness. Moreover, the orientation of a surface's normal vectors can be used to indicate the directionality of attributes. In the case of SYNBUILD-3D, semantics such as doors and windows are integrated into the wireframe surfaces. While building modeling and simulation software expects file formats such as .IDF or .GBXML, we represent each building wireframe in SYNBUILD-3D as a .JSON file which contains, among other things, a set of 3D coordinates and their respective adjacency matrix. As a result, SYNBUILD-3D can be readily integrated into machine learning pipelines without the need to conduct file format conversions.

To ensure the quality of the generated dataset, we enforce multiple sanity checks and constraints during the dataset creation process. First, procedurally generated building exteriors must exhibit footprint areas of at least 35 square meters since the footprint-conditioned floor plan generator tends to produce floor plans with only one or two distinct rooms for small footprint sizes. We also enforce that each floor plan must have at least three distinct room types. Second, to ensure that the floor plan vectorization identifies room boundaries correctly, we require each room polygon to have at least four corners. Third, the alignment loss defined in Equation 1 cannot exceed a threshold of 500. This threshold has been manually tuned by visually inspecting failure cases in the alignment process.

## 3.3 DATASET STATISTICS

SYNBUILD-3D consists of 103,233 semantically enriched 3D building wireframe models, corresponding floor plan images, and segmentation masks. To better understand the diversity and complexity of the 3D building wireframes in SYNBUILD-3D, Figure 3 and Table 2 illustrate the distribution of multiple building wireframe attributes, namely the distribution of node degrees as well as the number of nodes, edges, stories, rooms, windows, and doors across the individual buildings in the dataset.

While Figure 3a describes the distribution of the number of nodes, Figure 3b illustrates the distribution of the number of edges across all building wireframes. We can see that more than 99% of buildings in the dataset contain between 50 and 350 nodes, with a node median of 161 and a node mean of 162.48. We can also see that more than 99% of buildings exhibit a unique edge count between 50 and 450, with a median edge count of 210 and a mean edge count of 209.36 per building. To put these numbers into perspective, we can compare these statistics to Ramakrishnan et al. (2014), a molecule dataset which is widely adopted as a benchmark in molecular structure research and used for machine learning-based molecule generation tasks (Satorras et al., 2022; Xu et al., 2022; 2023). Similar to the 3D building wireframes in SYNBUILD-3D, each molecule in the

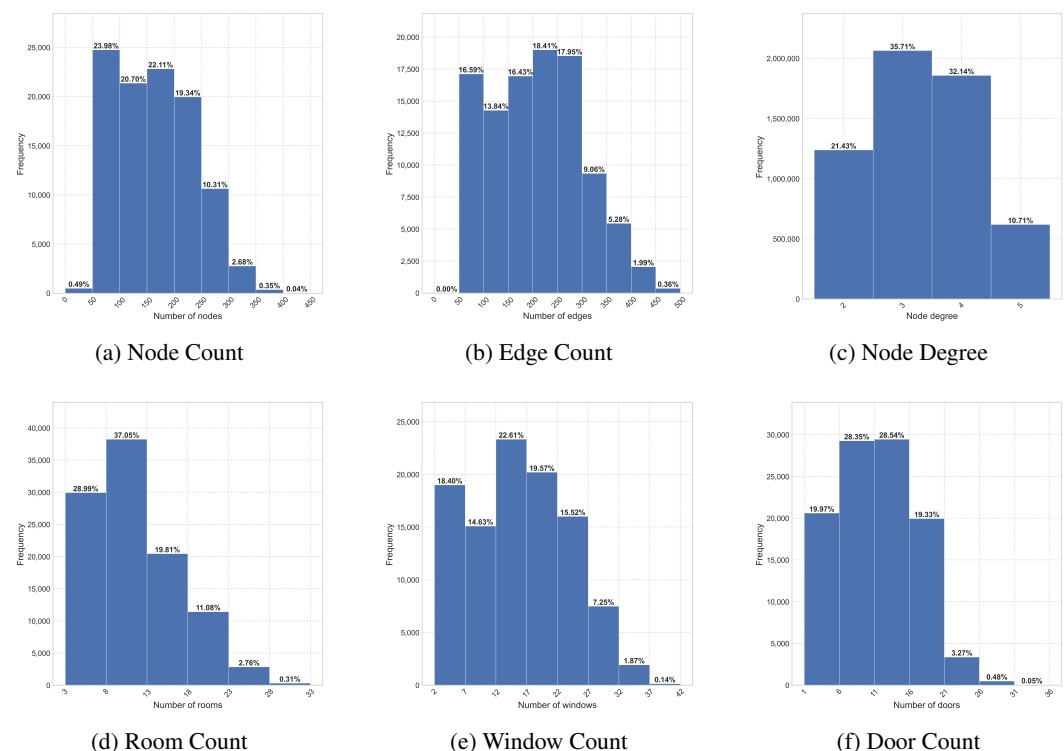

Figure 3: Feature distributions.

(a) Node Count (b) Edge Count (c) Node Degree

(d) Room Count (e) Window Count (f) Door Count

Ramakrishnan et al. (2014) is represented as a 3D graph consisting of individual atoms (nodes) and their respective bonds (edges). However, the molecular graphs contain only up to 29 atoms and, on average, around 19 edges. Hence, Figure 3a and 3b highlight the complexity and diversity of the generated 3D building wireframes in SYNBUILD-3D and their potential to support the development of algorithms for generating larger and more complex 3D graph structures.

Figure 3c describes the distribution of node degrees, i.e., the number of connected edges per node, across all building nodes in the dataset. Unlike Ma et al. (2024) where each building is split into separate wireframes for interior, exterior, and roof structures, resulting in node degrees no greater than three, our unified building wireframe representation leads to a significant share of nodes with a degree of four (32.14%) or five (10.71%), indicating an increase in wireframe complexity.

Lastly, Figures 3d–3f describe the distribution of the number of distinct rooms, windows, and doors in the dataset. We can also see that every building has at least three rooms.

In terms of room diversity, Table 3 shows that the three most common room types across all floor plans include living rooms (99.57%), kitchens (96.28%), and bathrooms (92.40%). The three least common room types include second bedrooms (34.51%), balconies (9.42%), and study rooms

Table 2: Building statistics.

| Feature | Mean ($\pm$ Std. Dev.) | Median |
|---|---|---|
| # Nodes | 162.48 ($\pm$ 72.39) | 161 |
| # Edges | 209.36 ($\pm$ 92.60) | 210 |
| Node Degree | 3.32 ($\pm$ 0.93) | 3 |
| # Stories | 3.00 ($\pm$ 1.41) | 3 |
| # Rooms | 11.12 ($\pm$ 5.79) | 12 |
| # Windows | 15.96 ($\pm$ 8.07) | 15 |
| # Doors | 11.24 ($\pm$ 5.95) | 12 |

Table 3: Distribution of room types.

| Room Type | Count | Percentage (%) |
|---|---|---|
| Living Room | 102,787 | 99.57 |
| Kitchen | 99,392 | 96.28 |
| Bathroom | 95,383 | 92.40 |
| Master Bedroom | 39,647 | 38.41 |
| Second Bedroom | 35,624 | 34.51 |
| Balcony | 9,727 | 9.42 |
| Study Room | 468 | 0.45 |

(0.45%). Note that each floor plan contains multiple rooms, which explains why the percentages do not add up to 100%.

# 4 Applications and Baselines

As a large multi-modal dataset of 3D building wireframes and their accompanying floor plan images, SYNBUILD-3D opens the door for multiple 3D generative modeling applications.

## 4.1 Unconditional 3D Building Generation

In a first step, SYNBUILD-3D can be used to develop, train, and validate generative algorithms that produce 3D building wireframes unconditionally. To do so, the developed algorithms would need to learn a distribution over the set of possible building wireframes and their associated semantics such that new 3D building wireframes can be sampled at inference time. While existing approaches for modeling 3D wireframe structures appear to converge to autoregressive (Ma et al., 2024; Nash et al., 2020) or diffusion-based techniques (Xu et al., 2022; 2023), both of which leverage a latent space obtained via different variants of graph-based auto-encoders, these approaches have not been tested on objects which exhibit a single wireframe representation for interior as well as exterior geometries and semantics. Hence, we argue that existing modeling approaches are not suitable as baselines for a dataset like SYNBUILD-3D. Instead, SYNBUILD-3D provides future research opportunities to extend or re-think existing 3D wireframe modeling approaches to a more complex set of objects.

## 4.2 Conditional 3D Building Generation

While generative models have made significant advances in text-to-3D and image-to-3D settings (Tang, 2022; Mildenhall et al., 2021; Kerbl et al., 2023), conditionally generating wireframes is an underexplored stream of research, gaining momentum particularly in the field of molecule generation (Xu et al., 2022; 2023). Taking advantage of the progress in the molecule generation field, SYNBUILD-3D can support the development of generative algorithms for 3D wireframes across various conditioning modalities. For example, by pairing floor plan images with corresponding 3D building wireframes, SYNBUILD-3D can, by default, support the development of image-to-wireframe models. In addition, leveraging the floor plan-derived semantic information for each building, SYNBUILD-3D can facilitate text-to-wireframe models. Moreover, since SYNBUILD-3D defines the complete 3D structure of every building in the dataset, it is possible to sample point clouds or wireframe subsets to train point cloud-to-wireframe and wireframe-to-wireframe models, e.g., for reconstructing full 3D building models from partial information. To conclude, SYNBUILD-3D adds value to the development of image-, text-, point cloud-, and wireframe-to-wireframe generative algorithms and across any combination of these conditioning modalities.

## 4.3 Proposed Metrics

To measure the quality of generated building wireframes, we propose to use metrics that capture their validity and uniqueness. Regarding wireframe validity, topology-aware losses are needed to ensure that the wireframe's connectivity and structure follow architectural design patterns in the built environment. Apart from well-established metrics such as Minimum Matching Distance (MMD), Coverage (COV), and 1-Nearest Neighbor (1-NN) based on Chamfer Distance (CD) and Earth Mover's Distance (EMD), another set of loss functions to measure the validity of angles and connections is needed. To holistically measure the integrity of generated 3D building wireframes, a graph edit distance-based loss to measure the cost of transforming a generated wireframe into its "closest" valid wireframe would be helpful. Following Ma et al. (2024), assessing the node degrees and their relative proportion across generated samples via KL divergence can indicate structural validity.

## 5 DISCUSSION AND LIMITATIONS

### 5.1 DISCUSSION.

We expect that SYNBUILD-3D will be a valuable dataset for the geometric deep learning community for the following reasons:

1. Designed in conjunction with experts in the building modeling and simulation field, SYNBUILD-3D is the first publicly available dataset of semantically enriched 3D building wireframes that adheres to industry conventions for building simulations. In particular, we model the building wireframes in SYNBUILD-3D to follow norms established by the widely adopted EnergyPlus simulation engine (National Renewable Energy Laboratory, 2017). Concretely, we model buildings as a single wireframe, i.e., each wall as a single 2D plane with no thickness. In contrast, Ma et al. (2024) models each wall as a set of two 2D planes, one for the interior surface and one for the exterior surface, an approach that is less favored by many practitioners in the 3D building simulation and modeling field.

2. Unlike Ma et al. (2024), where 3D building wireframes are constructed from a fixed set of 78,000 existing floor plans in the RPLAN dataset (Wu et al., 2019), SYNBUILD-3D and its associated pipeline provide a more flexible and modular approach for generating randomized 3D building wireframes at scale. This is because SYNBUILD-3D does not rely on a fixed set of floor plans to generate semantically enriched 3D building wireframes, but instead can automatically create new building geometries and floor plans at random. In addition, the modular structure of SYNBUILD-3D's data generation pipeline allows for the seamless integration of more sophisticated procedural generation engines (Pipeline Step 1) or footprint-conditioned floor plan generators (Pipeline Step 2) in the future. As a result, SYNBUILD-3D can easily be extended in the future.

3. To the best of our knowledge, SYNBUILD-3D is the first 3D building dataset that incorporates information on the position and size of doors and windows derived from floor plans.

4. Lastly, unlike Ma et al. (2024) where roof geometries for each floor plan are constructed ad-hoc and suffer from a lack of realism, SYNBUILD-3D leverages a principled algorithm to construct building-specific roofs and roof superstructures.

As a result, we foresee that SYNBUILD-3D can support the development of improved 3D generative algorithms which work across modalities, see Section 4.2, and learn to incorporate precise geometric and semantic relationships between individual structural elements.

### 5.2 LIMITATIONS.

While SYNBUILD-3D provides a novel approach and dataset for modeling semantically enriched 3D buildings at scale, we would like to discuss three existing limitations. First, the procedural generation engine that produces randomized building exteriors in Pipeline Step 1 only supports rectangular footprints (Biljecki et al., 2016). As a result, SYNBUILD-3D does not contain building wireframes with more complex footprint geometries, even though the footprint-conditioned floor plan generator in Pipeline Step 2 can handle rectangular and non-rectangular footprints (Wu et al., 2019). To mitigate this limitation, we have designed SYNBUILD-3D's pipeline in a modular fashion so that more complex procedural generation engines for building exteriors can be integrated seamlessly as they become available. Second, as a simplifying assumption, we assume that the floor plan of a given building does not change across different floors. This simplification can be mitigated in the future by generating a unique floor plan for each building floor while incorporating accessibility constraints. Third, roof superstructures such as windows and dormers are not considered during the floor plan generation step, which, in rare cases, can lead to roof windows that overlap with underlying walls.

## 6 CONCLUSION

We presented SYNBUILD-3D, a novel, multi-modal, and synthetic dataset of more than 100,000 semantically enriched 3D building wireframes with AI-generated floor plans. SYNBUILD-3D has

been designed with and validated by building modeling and simulation experts. It offers detailed information on interior and exterior building geometries and rich semantics, including information obtained from floor plans such as room types and the size and position of doors and windows. Moreover, since each building wireframe comes with its corresponding floor plan image and segmentation mask, SYNBUILD-3D can support unconditional as well as conditional generation tasks across a wide set of modalities. Lastly, SYNBUILD-3D benefits from a modular and scalable generation process that automatically produces a wide variety of randomized 3D buildings and corresponding floor plan images. As a result, our dataset of 103,233 buildings can easily be extended in the future and benefit from the integration of more advanced procedural generation engines and footprint-conditioned floor plan generators as they become available.

By releasing SYNBUILD-3D along with its codebase, the geometric deep learning community now has access to a dataset which should facilitate the development of improved generative algorithms for complex 3D objects, ranging from modeling exact geometric and semantic relationships to a variety of cross-modal building wireframe generation tasks.

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
