# OpenReview forum: "SYNBUILD-3D: A multi-modal synthetic dataset of over 100,000 semantically enriched 3D building wireframes with AI-generated floor plans"
_ICLR.cc/2025/Conference — ICLR 2025 Conference Withdrawn Submission_

### Official Review · Reviewer_u7QK · 2024-10-27

**Soundness:** 2
**Presentation:** 1
**Contribution:** 2
**Rating:** 1
**Confidence:** 4

**Summary:**

The paper introduces SYNBUILD-3D, a large, multimodal dataset comprising over 100,000 3D building wireframes and their corresponding floor plan images, aimed at improving the modeling of geometric and semantic relationships in 3D spaces. Developed in collaboration with experts in building modeling and simulation, SYNBUILD-3D offers detailed interior and exterior geometries, including the precise locations and sizes of doors and windows, making it unique among existing datasets. This dataset is intended to support the geometric deep learning community in both conditional and unconditional 3D building generation tasks, addressing a significant gap in the availability of high-quality 3D data.

**Strengths:**

N/A

**Weaknesses:**

1. The SYNBUILD-3D dataset is slightly larger than the 3D House Wireframe dataset, but since it is synthetic, I didn't notice any significant improvements compared to the 3D House Wireframe dataset.
2. The paper primarily presents this synthetic dataset without benchmarking against baseline methods.
3. There is no preview for the proposed dataset. I cannot evaluate the quality of the proposed dataset.
4. The paper falls short of the required page count for ICLR submissions, containing only 8-9 pages instead of the mandated 9-10 pages. **Submitting this unfinished paper to ICLR in hopes of getting lucky is a complete waste of both the reviewers' and the AC's time.**

**Questions:**

Please refer to the weaknesses.

---

### Official Review · Reviewer_1RdJ · 2024-10-28

**Soundness:** 2
**Presentation:** 1
**Contribution:** 2
**Rating:** 3
**Confidence:** 4

**Summary:**

This paper presents SYNBUILD-3D, a large-scale dataset developed to address the challenges of 3D building generation using graph-based representations. Existing datasets are often limited, exhibiting proprietary, simplistic, and focusing on exterior-only structures, the authors introduce SYNBUILD-3D to support the creation of high-quality 3D building models enriched with complex geometric and semantic details. The dataset includes over 100k 3D building wireframes, each accompanied by floor plan images and segmentation masks, enabling various generative tasks in both unconditional and conditional settings.
The SYNBUILD-3D dataset was created through a multi-step pipeline designed for architectural realism. This process begins with the procedural generation of building exteriors, followed by AI-based floor plan generation conditioned on each building's footprint. To ensure quality, SYNBUILD-3D integrates various technical validations and constraints recommended by building modeling experts.

**Strengths:**

This paper introduces SYNBUILD-3D, a pioneering dataset in computer vision specifically designed for 3D building generation, addressing the gap in large-scale, semantically enriched architectural data.
+ Novel contribution to 3D building data: SYNBUILD-3D fills a critical gap in computer vision and architecture by providing a large-scale dataset specifically designed for 3D building generation.
+ Multi-modal dataset format: By pairing 3D wireframe models with floor plan images and segmentation masks, the dataset can potentially enable generation tasks across 2D and 3D domains, and supports a broader range of applications, from architectural design to spatial planning.
+ Scalability and flexibility: The pipeline is capable of generating large-scale data and is also flexible enough to integrate new generative modules as advancements in building modeling emerge.
+ Expert quality validation: Quality control in SYNBUILD-3D is well-executed, with validation checks recommended by experts in building modeling.

**Weaknesses:**

- Diversity of the dataset: While SYNBUILD-3D uses procedural generation for creating building models, the lack of real-world architectural diversity could limit the dataset’s applicability for modeling highly variable or culturally specific structures (e.g. not all buildings in the world are residential buildings, what about office buildings? Also modern buildings and historical buildings can have very different layouts). The examples illustrated in the paper also shows limited diversity.
- Limited structural complexity: The dataset enforces minimum footprint areas and alignment thresholds to ensure model consistency, which may exclude smaller, intricate structures, e.g. unusual room shapes, densely packed spaces. There is a gap between this representation and real-world scenarios.
- Demonstration: the demonstration quality of this paper is sub-par, with limited number of examples, limited variation of the building and without supplimentary. Only 4 examples are given in the paper, all demonstrating simple and similar structures (extruded from a floorplan). Feature distribution in Fig3. is not a direct evidence for data diversity.
- Downstream application motivation is weak: authors mentioned unconditional (e.g. layout generation) and conditional 3D building generation (point cloud to wireframe, and wireframe to wireframe, also image to wireframe). However, the proposed dataset does not contain paired point cloud data or rendering of the room (except floormap). The application seems quite narrow and less straightforward.

**Questions:**

- Is it possible to expand beyond basic floorplan and wireframe generation, which only indicate room locations, doors, and windows, to support more complex and intricate indoor scene layouts?

- The choice to exclude existing works as baselines is unclear. The authors should benchmark established algorithms on their dataset; if these models were trained on other datasets, they could be retrained with the new dataset. Testing or fine-tuning large language models (LLMs) on generation tasks would also be straightforward.

- Wireframe generation can be approached similarly to tasks like layout and mesh generation, albeit with unique constraints. For example, HouseGan and HouseGan++ address room floorplan generation; if their input bubble diagrams were modified to include walls, doors, and windows, they would serve as plausible baselines. BlockPlanner, which tackles city layout generation with vectorized representations similar to wireframes, could also be adapted for this task. From a mesh perspective, MeshGPT could similarly be adapted for wireframe generation. LLMs also seem well-suited, as LayoutGPT has shown effectiveness in image and room layout generation. The authors should discuss relevant approaches and potential modifications to these models for wireframe generation in this context.

---

### Official Review · Reviewer_w6oE · 2024-11-08

**Soundness:** 2
**Presentation:** 2
**Contribution:** 2
**Rating:** 3
**Confidence:** 4

**Summary:**

This paper introduces SYNBUILD-3D, a new multi-modal dataset designed to address the scarcity of 3D datasets, particularly for buildings. The dataset includes 100,000 3D wireframe models of buildings paired with corresponding floor plan images. Each 3D model captures both the exterior and interior of a building, annotated with room types and structural details, including the position and dimensions of windows and doors.

To build this dataset, the process starts by generating random building exteriors using an off-the-shelf procedural generation engine. The building's footprint is then used to guide a separate floor plan generator specialized in residential layouts, producing a compatible floor plan for each structure. This floor plan is vectorized and annotated with room labels and positions and sizes of doors and windows. Following this, an alignment optimization step is applied to ensure that the floor plan aligns accurately with the building's footprint. The aligned floor plan is then extruded to the volume of the building.

The paper also discusses potential applications for SYNBUILD-3D in tasks such as unconditional and conditional 3D building generation.

**Strengths:**

- A large-scale 3D building dataset specifically designed to support generative methods in the 3D domain.
- In addition to 3D wireframe models that capture both exterior and interior geometry, the dataset includes annotated floor plans that offer further utility for generative applications.

**Weaknesses:**

- The dataset focuses exclusively on residential buildings, as it leverages an off-the-shelf floor plan generator specialized for this type. This is critical, as datasets intended for generative approaches must not only be large-scale but also provide a diverse set of data within a specific domain.
- The paper does not include baseline results demonstrating the dataset’s effectiveness for 3D building generation. While it provides a brief discussion on potential methods, it is crucial to include quantitative and qualitative evaluations of these methods (or their adaptations) on the proposed dataset. Additionally, incorporating a graph-based baseline that leverages the dataset’s inherent graph-like structure would be beneficial, as this could illustrate how such a representation supports the outlined generative tasks.
- Directly transferring room type labels and door and window geometries from the generated floor plans to annotate the 3D wireframes results in multistorey buildings with unrealistic, repetitive interiors and exteriors. For example, it is uncommon in a three-storey building to have a kitchen on each floor, stacked one above the other.

**Questions:**

The initial building polygons are generated using the Random3Dcity procedural modeling engine. Which level of detail (LOD) was selected for the dataset? Wouldn’t it be more beneficial to create multi-LOD buildings, where each sample includes a hierarchy of buildings, in a coarse-to-fine manner?

---

### Official Review · Reviewer_rc4c · 2024-11-08

**Soundness:** 2
**Presentation:** 2
**Contribution:** 3
**Rating:** 5
**Confidence:** 3

**Summary:**

The paper introduces SYNBUILD-3D, a large, synthetic dataset of over 100,000 semantically enriched 3D building wireframes, each accompanied by corresponding AI generated floor plan images and segmentation masks. Designed to support the development of generative AI models for 3D building design, the dataset offers a unified and detailed representation of both building interiors and exteriors. It provides rich semantic information, including room types and the positions and sizes of doors and windows. The dataset generation pipeline combines procedural generation with an AI-based floor plan generator, and extensive validation and attribute analysis support its quality and robustness for machine learning applications.

**Strengths:**

1. SYNBUILD-3D fills a significant gap by providing detailed interior and exterior building structures with semantic annotations, which is crucial for training AI models capable of generating realistic and complex 3D buildings.

2. The dataset benefits from a robust generation pipeline that combines procedural generation with an AI-based floor plan generator. The extensive validation and detailed attribute analysis ensure the dataset's quality and make it suitable for machine learning applications.

**Weaknesses:**

1. Lack of Detail on the AI-Based Floor Plan Generator:
The paper does not provide sufficient information about the AI-based floor plan generator, which is a key component of the dataset. This lack of detail impacts transparency, reproducibility, and the ability to assess any limitations in the floor plan generation process.

2. Limited Architectural Diversity Due to Rectangular Footprints:
The procedural generation engine only supports rectangular building footprints, which may limit the architectural diversity and realism of the dataset. Real-world architectural applications often require varied building geometries.

3. Lack of Evaluation with Baseline Models:
The paper does not present any baseline results or example models trained on SYNBUILD-3D, making it difficult to assess the dataset's immediate effectiveness for advancing generative 3D building research. Providing benchmark results would strengthen the paper's claims about the dataset's utility.

**Questions:**

Including a detailed description of the AI-based floor plan generator, with information on its structure, training data, and any design constraints, would improve the dataset’s transparency. Adding baseline results or examples of model pipelines would further demonstrate SYNBUILD-3D’s applicability. Addressing footprint diversity and multistory design variation would enhance the dataset’s relevance to real-world architectural tasks.

---

### Note · Authors · 2024-11-22

**Comment:**

Dear reviewers,

Thank you for your time and feedback. We will work on improving the manuscript based on your reviews, but believe that the discussed changes require more time than we have in this rebuttal period, hence we decided to withdraw.

**Withdrawal Confirmation:**

I have read and agree with the venue's withdrawal policy on behalf of myself and my co-authors.